

# From geoscientific 'matters of fact' to societal 'matters of concern': a transdisciplinary training approach to communicating earthquake risk in Istanbul (Turkey)

J. Ickert[1] and I. S. Stewart[2]

[1,2] Sustainable Earth Institute, Plymouth University, Plymouth PL4 8AA, UK.

*Correspondence to*: J. Ickert (johanna.ickert@plymouth.ac.uk)

**Abstract.** An important paradox of hazard communication is that the more effectively a potential physical threat is made public by the scientist, the more readily the scientific message becomes normalised into the daily discourses of ordinary life. As a result, a heightened risk awareness does not necessarily motivate personal or collective preparedness. If geoscientists are to help at-risk communities adopt meaningful measures to protect themselves, new strategies are needed for public communication and community engagement. This paper outlines an attempt to develop a transdisciplinary approach to train geoscientists, using early career researchers in an EU integrated training network studying tectonic processes and geohazards in Turkey. An urban field visit to seismically-vulnerable neighbourhoods in Istanbul allowed a group of young geoscience researchers to meet with local residents facing the seismic threat. Those meetings exposed the complex social, political and cultural concerns among Istanbul's at-risk urban communities. These concerns were used to provoke subsequent roundtable discussions among the group of geoscientists about the roles and responsibilities of communicating hazard information to the public. Through the direct testimony of local residents and geoscientists, we explore the form that new strategies for public communication and community engagement might take.

## 1 Introduction

Hazard scientists rarely meet the people that are actually 'at risk' - those in communities prone to natural threats. When they do, scientists generally find that those living in the shadow of disaster view an impending threat in ways very different to that envisaged by the specialist, whose outlook is steeped in probabilistic or deterministic thinking about the chances or impacts of an extreme event. Unfettered by the technical prognosis for a particular hazard scenario, ordinary citizens instead embed scientific concerns about the likelihood of a natural calamity into the broader social, economic and political stress field that shapes their day-to-day lives. The projected earthquake, volcanic eruption or flood event feeds into community conversations about topics such as ongoing social transformations, local arguments over economic development plans, and political debates about corporate corruption and civic trust. Indeed, an important paradox of hazard communication is that the more



effectively a potential physical threat is made public by the scientist, the more readily the scientific message becomes normalised into the complex, chaotic and contested discourses of daily life.

Such a situation confronts the issue of earthquake preparedness in Istanbul. The geoscientific consensus is that this city of 13.5 million inhabitants is facing a major earthquake threat in the coming decades (Parsons et al., 2000: Bohnhoff et al., 2013). The destructive earthquakes of August and November 1999 to the east of the city highlighted that lethal potential of the seismic threat (Özerdem, 1999), and the intervening years has built up a considerable body of science concerning future disaster scenarios (e.g. Barka, 1999; Okay et al., 2000; Pichon et al., 2001; Armijo et al., 2005; Ansal et al., 2009; Erdik et al., 2011). In response, geologists and engineers have been involved in city-wide earthquake preparedness measures, mainly focused on improving the resilience of the city's largely inadequate building stock. A loss-estimation study carried out for Istanbul after the 1999 Kocaeli earthquake (JICA & IMU 2002) revealed that, under a scenario earthquake of magnitude 7.5 along the Marmara Sea segment of the North Anatolian Fault, over 50,000 buildings could expect to be heavily damaged or collapse. Despite a recognition that "seismic risk in the buildings in Istanbul is mostly dominated by building vulnerability, not hazard" (Yakut et al. 2012, p.1545), there is widespread distrust of Istanbul's retrofitting and reconstruction measures even among residents of some of the city's most at-risk quarters (Green, 2008; Islam, 2010; Karaman, 2013; Kuyucu, 2014; Özkan and Özcevik, 2015).

The roots of this distrust go deep into the Turkish psyche. An inter-comparison of populations living in seismic earthquake-prone areas in Japan, USA and Turkey revealed that especially strong and varied emotions permeate Turkish earthquake perceptions and attitudes (Joffe et al., 2013). The direct experiences with the 1999 earthquakes had heightened feelings of worry, fear and anxiety, but in addition there were strong expressions of corruption and incompetence of politicians, civil servants, planning regulators and the construction industry. According to the study, discussion of corruption accompanied expressions of lowered self-esteem, and two-thirds of Turkish respondents lamented a 'demise of identity', with responses to earthquake risk permeated by the widespread belief that the character and moral fibre of the country was weak and ineffective. For many participants, it was this endemic corruption, greed and selfishness that was seen to produce vulnerable cities and buildings, and which produced a heightened fatalism and weakened sense of control self-efficacy (Joffe et al., 2013). The result was that despite a substantial awareness of the earthquake risk, the Turkish respondents were far less likely than their US or Japanese counterparts to adopt seismic adjustment measures. As a consequence, while overall awareness of Istanbul's inhabitants for an impending seismic threat is high, there remains an acute lack of civic preparedness (Özerdem, 1999; Eraybar et al., 2009; Erdik, 2014).

The contested nature of Istanbul's seismic preparedness exemplifies a general view emerging from disaster risk reduction research. This view stresses the necessity for hazard practitioners to pay more attention to the social and cultural dimension of risk, and to analyse how, and if, adaptation and mitigation measures integrate local concerns (Krüger et al., 2015; Drake et al., 2014; Voss et al., 2014). For the geoscientist charged with a responsibility to communicate the earthquake hazard,





addressing the social and cultural dimensions of seismic risk is problematic. Most hazard scientists are trained in the physics of natural processes and practised in intricate risk assessment procedures, but not in the nuances of political science or cultural theory, nor the sociology and psychology of human relations. For that reason, most geoscientists would regard it as beyond their realm and remit to confront the messy reality of how natural threats are translated and perceived by an at-risk

community. Instead, the standard communication approach relies on geoscientists delivering clear and transparent messages that report the empirical evidence of sound technical science.

The difficulty is however, that for more than a decade social science studies indicate that there is little or no correlation between the provision of scientific information about geohazards and risks and the adaptive changes in individual or community behaviour that would reduce risk (Palm & Hodgson, 1992; Solberg et al., 2010; Joffe et al., 2013; Renn et al.,

2013; Wachinger et al., 2013; Krüger et al., 2015). In other words, as is evident in Istanbul, a high risk awareness does not necessarily motivate personal or collective preparedness. The key message from social science is that simply getting the scientific message across to a vulnerable public is not enough. If geoscientists are to truly help an at-risk population adopt meaningful measures to protect itself, then arguably they need to design new strategies for public communication and community engagement.

In this paper, we attempt to address this 'deficit gap' between what geoscientists want to tell at-risk communities and what those communities want to hear from the scientific experts. We do so by reporting on a communication workshop that directly provoked a group of early-career geoscientists with the sharp focus of the local politicized nature of seismic preparedness in Istanbul. The aim was to challenge the geoscientists about their role as 'communicators', and expose what are the essential geoscience messages that need to be publicly conveyed. With that objective, in the following sections we

first introduce the geoscientists, and then outline the political and social context in which Istanbul's seismic risk controversy is embedded. The paper then documents the responses of the geoscientists to the communication problem they have been confronted with, and follow that up with recommendations that emerge from group about establishing new strategies for geoscience communication in general and community-centred earthquake education initiatives in particular.

**2 The participants**

The geoscientists in question were a group of doctoral and postdoctoral researchers engaged in a Marie Curie Integrated Training Network on 'Anatolian pLateau climatE and Tectonic hazards' (ALErT). ALErT's emphasis on natural hazards – principally earthquakes, landslides and flooding – means that in addition to receiving training in advanced methods of geoscience data acquisition and field investigation, the young geoscientists are expected to develop expertise in effective

science communication, both to decision-makers and the wider public. Indeed, as the ALErT proposal states:

*'Delivering basic information on hazards to those who are most at risk is recognized as a fundamental and*



*persistent weakness in disaster risk reduction programs worldwide. Addressing this deficiency requires a combination of 'top-down' technocratic approaches, in which scientific expertise is communicated down formal decision-making chains of command, and 'bottom-up' community-based approaches, in which that expertise feeds into local educational initiatives to build resilience among those at risk.'*

The scientific backgrounds of the ALErT group are drawn from seismology and geophysics, tectonics, geology and geography, hydrology and palaeoclimate, and spatial analysis and geo-statistics. Such diverse geoscience specialisms present difficulties for overcoming technical jargon and methodological barriers emerging from a disparate suite of analytical techniques and research approaches. To compound the difficulty, many of the researchers are concerned with very different

time frames, some tracking processes that operate over several millions of years, other studying phenomena encountered in recent human memory. Thus, in a project that juxtaposes a researcher collecting microfossil samples from 10-million year old sediment deposits in montane basins with one that is acquiring offshore seismic reflection data from marine survey vessels and another that is creating a regional database of modern hydrological flood events, it is easy to perceive that the individuals involved have little scientific overlap and that merging various viewpoints into a common communication

strategy faces major difficulties. Such a mixed expertise and such a fragmented knowledge base means that the researchers have to spend a considerable amount of time trying to understand each other before attempting to communicate their findings to an even less knowing public (Scherer and Renn, 2015).

Another challenge for this international group of early researchers emerges from their own social and cultural diversity. Around half of the 12 researchers in the ALErT team come from Turkey, with the remainder from Germany, Netherlands,

Spain, Sweden, and USA; only the US participant has English as a first language, and across the group the level of proficiency and confidence in written and spoken English was variable. The individual researchers are now almost all based at institutions or organisations outside their home country, in academic settings that are sometimes very different to those in which they undertook their initial studies. Some are products of traditional, overtly proscriptive approaches to formal geoscience training, now transplanted into more student-centred or outward-facing learning environments. For others, the

reverse is true. The result is that the group is an amalgam of academic cultures in which the exposure to generic communication skills varies markedly from individual to individual and the impetus to engage with external publics is inconsistent. A common element, however, was that none had received any formal academic training in science communication.

As a group, the potential geo-communicators within the ALErT consortium constitute a highly specialised and academically

disparate collective of researchers, essentially unpracticed in science communication, applying their technical expertise in a cultural setting that most of the participants are unfamiliar with. In such a context, standard graduate training in generic science communication principles and practices was considered unlikely to overcome the various cultural and academic



barriers to conveying their scientific expertise to those that need it. To counter this, a 'level playing field' was invoked in the form of directly confronting the communication asperities presented by the acute problem of Istanbul's seismic threat.

## 3 The context: Istanbul's political earthquake

The destructive Kocaeli and Duzce earthquakes of August and November 1999, although located east of the city, brought

home to many Istanbul residents the likelihood of a future direct seismic strike on the metropolitan area (Özerdem, 1999). The two earthquakes led to a region-wide disaster that caused close to 20,000 deaths and over 54,000 damaged buildings (Erdik, 1999). Geological investigations have revealed that the principal seismic threat comes from an 'earthquake gap' in the North Anatolian Fault immediately south of the city (Stein et al., 1997; Parsons et al., 2000; Armijo, 2005; Bohnhoff et al. 2013), but the lethality of any large (M>7) earthquake triggered beneath the waters of the Marmara Sea largely arises

within the city itself. When the earthquakes struck in 1999, the majority of housing in Istanbul did not even meet minimum building standards specified in the earthquake design codes introduced in 1944, and updated in 1953, 1968, 1975, and 1998 (Soyluk and Harmankaya, 2012). Based on statistics from the 1999 events, it is estimated that the multi-storey reinforced concrete buildings that dominate modern Turkey are 10 times more vulnerable to earthquakes than similar buildings in California exposed to the same level of hazard (Erdik and Aydinoglu, 2002). Accordingly, 30-40% of Istanbul's building

stock is considered to be at risk (Erdik and Durukal, 2008; Bugra, 1998).

The acute seismic vulnerability of the Istanbul's built environment is a direct product of its rapid unauthorized urban growth from 1930, when this capital of the Ottoman Empire housed 800,000 residents, to 2000 when its population surpassed 10 million people (Green, 2008). Facilitating this rampant unplanned industrialisation and urbanisation was the proliferation of Istanbul's informal housing districts, locally called 'gecekondu' neighbourhoods. These squatter districts emerged during the

onset of massive rural-urban migration that started in the 1940s (Bugra, 1998; Green, 2008). The districts are dominated by low-quality, sub-standard buildings, erected within a short time (the term "gecekondu" is Turkish for "built over night") and typically without any professional consultation of planners or architects (Bugra, 1998; Green, 2008). The casual nature of the construction means that this self-built housing is especially vulnerable to earthquakes, and its intrinsic vulnerability was heightened further in the 1980s when a series of amnesty laws legalized a large percentage of the informal building stock. As

a result, many existing 1-2 storey gecekondus were extended into "post-gecekondu" settlements with 3 or more storeys (Esen et al., 2005).

In an attempt to strengthen the seismic safety of the city, in the mid-2000s, Istanbul's civic authorities introduced an ambitious programme of "Urban Transformation" projects, also known as "Urban Renewal", during which many gecekondu districts underwent large-scale retrofitting and reconstruction. Istanbul's urban transformation projects have been

accompanied by major public protests, especially within gecekondu districts. Despite broad societal support for the necessity of risk reduction efforts, the main popular objections relate to socioeconomic trade-offs, negative environmental impacts,



triggered gentrification processes and democratic deficits, especially in the lack of citizen participation (Islam, 2010; Adalani, 2013; Turam, 2013; Balamir, 2013; European Commission, 2013/2014; Angell, 2014; Özkan and Özcevik, 2015). Prevailing divides and entrenchments between the local communities and civic authorities in charge of the mitigation measures were intensified by the perception of a strongly hierarchical disaster management structure in Turkey. This

organizational structure lacks formal mechanisms to facilitate interchange between academic scientists and the general public, and more critically is devoid of participatory decision-making with at-risk local communities: shared 'platforms', consensual implementation of projects, devolved forms of governance, the involvement of resident groups in the identification of local vulnerabilities (Balamir, 2013).

The result is a disconnect between scientific pronouncements about serious hazard threats and apparently low levels of

precautionary action at the individual and community level (Taylan, 2015). And it is against this contested and highly political backdrop that earthquake scientists, geologists and engineers are compelled to communicate.

## 4 The field excursions: insights from Istanbul's seismic suburbs

In late May 2015, under the guidance of local urban historian Orhan Esen, the 14 geoscience researchers undertook a half-day 'field' visit to the gecekondu districts of Zeytinburnu and Okmeydanı. The visit gave them a first-hand picture of the

building stock of both neighbourhoods and provided the opportunity to meet several inhabitants and community representatives. In the framework of street interviews initiated by the researchers in Zeytinburnu and an extended roundtable discussion with Ali Çetkin, chairman of the Okmeydanı-based neighbourhood association "Okmeydanı Çevre Koruma ve Güzelleştirme Derneği" the geoscientists were confronted firsthand with the community perceptions of seismic risk communication in both districts.

The following section outlines three key aspects that emerged out of conversations between the geoscientists and inhabitants. These summarize local perspectives on the mitigation process relevant to consider when developing communication strategies for (and with) for at-risk communities.

### 4.1 Side-effects of urban transformation on disaster preparedness

The dramatic transformation of the gecekondu districts were noted by all of our interview partners, who acknowledged the

seismic threat as the main official argument for the urban renewal projects.

> *"We have been living here for 30 years. This used to be a football field, then there was an urban transformation*
> *process, so people were being taken to these new buildings that are safer for earthquakes. It was an empty area,*
> *it was just a sport area before." (Sedat, Zeytinburnu)*




Although living conditions in the new Zeytinburnu apartment blocks were regarded as now being "comparable to European standards", the construction of large multi-storey apartment blocks attracting additional tenants marked a worrying increase of population density in the high-risk district.

> *"By this kind of market-driven risk mitigation, you have to raise the density. Because the financing goes through the market, not through public funding. The increasing of the density is in clear contradiction to the requirements of earthquake building codes." (Orhan Esen, Okmeydanı)*

Moreover, the construction of new high-rise apartment blocks, shopping-malls, private car parks etc. was criticised for taking over previous open spaces that would be needed in the post-disaster phase for evacuation.

> *"For the rescue just after an earthquake, we would need free spaces. So obviously the government doesn't take the risk seriously." (Ali, Okmeydanı)*

Perhaps more significantly, Zeytinburnu residents lamented the increased anonymity brought by the large influx of "new people that moved into the project", a product of the engineered gentrification processes. Despite an agreement on the necessity of physical risk mitigation measures, and an appreciation of modernized, earthquake-proof apartments, residents reported unintended social side-effects of a risk reduction strategy focused mostly on physical measures.

> *"The residents are supposed to solely stay within this compound, this gated community. So you have your social club inside, you have your swimming pool inside, your sports facilities, a kindergarten and so on. Which you obviously didn't have in the former 'mahalle', in the former quarter, which is now being pulled down piece-by-piece. But interesting is that our interview partner said that although they are living in the compound, they prefer still going to the old café which will now also be pulled down. He plays cards there with his companions. So obviously he sees this environment much more friendly. Obviously the new compound that has been built lacks some quality." (Orhan Esen, Zeytinburnu)*

In Okmeydanı, a loss of cultural heritage was described, with an impact on the cultural identity of the neighbourhood:

> *"They destroyed all of what was here. This is a former pastoral bath house, it is built into the foundations of the new building,(...) they just pulled it down to build a minaret, which has nothing to do with the old one. There was an*



*open air prayer space, it has also been pulled down and reassembled. It has nothing to do with the old one. These are just disneyland fakes of the originals." (Ali, Okmeydanı)*

The modernization of the district also goes along with a fear of the inhabitants to be evicted. According to Orhan Esen, only 30-40% of the former inhabitants can afford to live in the new projects:

*"It is a working class area. Most of the people cannot afford such standards. They never paid rent, but they are not qualified for the job market either. They still work as unqualified labourers. Whenever they move in the new compound, they cannot afford the new lifestyle there, they cannot keep up the payments. Here in Zeytinburnu, which is quite a well-off middle class community of Istanbul, it is like one third that could make it into the new project. In no case you can expect more than 30-40% of the former inhabitants to live in the new projects."*
*(Orhan Esen, Zeytinburnu)*

Taking into account that the Urban Renewal has very particular consequences on community resilience, the fragmentation and dissolution of community resilience is a theme apparent already in previous attitudinal surveys, with Joffe et al. (2010) noting the heightened feelings of isolation, despair and sadness encountered among Turkish respondents when it comes to seismic risk adjustment.

### 4.2 Lack of co-determination, cooperation and transparency

A persistent complaint among interviewees related to a lack of citizen participation in the risk mitigation process, highlighting few, if any, established forums for science-public exchange, and an absence of contributions of local communities in the planning process and in regeneration activities.

*"37 houses were pulled down in the year 2005. They said 'we are going to make you a park.' But they didn't make any official announcements. The owners of the 37 houses never received any official documents." (Ali, Okmeydanı)*

Residents stated that "there weren't any plans made in cooperation with the public". They found out about the mitigation plans "through the actual demolitions" or through occasional municipality-based "events" which were seen as showcases aiming to secure public acceptance for mitigation measures rather than as opportunities for open dialogue. Residents criticized that during these "events" the organizers "talked the entire time and wouldn't allow us to ask critical questions". These experiences led to a "growing distrust" in the authorities responsible for the mitigation process, who were criticized for "mostly building for themselves and their profit", and not for the safety of the residents.



Informal comments expressed on the ground in Zeytinburnu and Okmeydanı endorse the findings of Green (2008) and Joffe et al. (2010). These document that widespread complaints of corruption in the political sphere as well as in the construction industry, the commercialisation of urban development and a marginalisation of the inhabitants exposed by the seismic risk feeds a growing distrust in the quality of seismic safety of the newly built apartments and nourishes feelings of fatalism.

> *Ali: Three years after, they demolished the park, they built (...) an exclusive club for archery. (...) But even this is just make up. Because their real concern is converting that whole area into a shopping mall. They have already built four elevator shafts. It is prepared for building up."*
>
> *Researcher 4: Do you feel prepared for an earthquake?*

> *Ali: There is no preparation, that is for sure. But do you think that there is any preparation in any other districts other than Okmeydanı? (...) We don't believe this government, because if they just built this exclusive archery club and declare this as a kind of a measurement, vis-a-vis the earthquake risk, what does this have to do with earthquake mitigation? They just built things for themselves. Within their whole ideological context they built an exclusive club. It doesn't have to do anything with an earthquake. So what gives us the reason to believe in anything they do about the earthquake?*

## 4.3 Deficiencies of seismic risk communication

Following on from this perceived lack of co-determination, cooperation and transparency, residents also expressed a lack of trust in most mediated geoscience. This sentiment was based on the observation that the seismic risk issue was seen as being used as an instrument for the real-estate and insurance industry.

> *"We are not informed at all. What we believe is that the earthquake is just used as alibi, as an excuse, as a pretext. The term earthquake doesn't point to the real thing. The real thing is that they want to acquire this very precious land here." (Ali, Okmeydanı)*

25  Another argument was that scientific research results and conclusions were viewed to be frequently and widely misrepresented in the media coverage, usually by heightening the consequences and not sufficiently supporting the preparedness of the inhabitants, who felt left "misinformed". Other complaints related to the poor accessibility and usability of scientific information given out by public institutions, which were often deemed incomprehensible, not targeted at or written for ordinary citizens. Inhabitants also expressed a confusion about the role and responsibilities of the institutions in charge of risk mitigation. In Okmeydanı there were specific complaints about a lack of transparency and scientific evidence
30  for the municipalities high-risk designation of the neighbourhood.



*Researcher 3: These houses are safer than others?*

*Orhan: Supposedly, officially. By the very official discourse they are.*

*Researcher 5: Is there also more safety during earthquakes?*

*Orhan: I cannot say. The official justification for this project is that this mitigates the risk.(...)*

*Researcher 10: Concerning the kind of data for the red areas, [designating the seismic high risk areas]- what kind of data is it?*

*Orhan: There is no data! It is not data, it is something else. (...) All red areas that are designated as urban transformation areas for the sake of risk mitigation are areas where some private developers showed interest for whatever reason. There is no scientific criteria, nothing. If there is a group of developers that show an interest in*
*transforming that particular informal housing area, that area is transformed into a disaster risk area.*

Although residents expressed their concerns about the quality of seismic risk information given out by the municipality and their contractors, they expressed their trust in independent geoscientists.

*Researcher 3: It seems as if you don't have a lot of faith in the government, but do you have a lot of faith in scientists?*

*Ali: Of course, why shouldn't we trust scientists? A major reason why we don't trust the government is that they already founded a development company to market our neighbourhood. This is already part of the official newspapers around Turkey. It is not that we don't trust the government out of ideology, but just by the very facts we see.*

*Researcher 10: But the science comes mainly from the universities, and the universities are mainly driven by the government. So it is a paradox. You don't trust the government but trust science which comes from the government.*

*Ali: Of course there are differences between universities and universities, scientists and scientists. Of course we are aware of that, but there is also something that we can call "common sense". And maybe we are not geologists, but*
*we also have our education in different fields. We are experts in our field. That allows us to judge in a proper way. Of course we also have our interests. As citizens, we can kind of measure our interests and our expertise in our field. And the common sense will help us to differentiate between scientists and scientists (...)*

In the past, the Okmeydanı neighbourhood organization had already tried to access scientifically valid and understandable
information, and consulted geoscientists from universities. Yet, the exchanges were described as problematic. Factors named were problems to understand the scientific terminology and difficulties to extract relevant knowledge for their specific locale. Also, the independence of the geoscientists were seen in as restricted, as they "didn't want to give out written reports"




and "didn't want their names to be publicly mentioned". After these unsatisfying attempts to gain sufficiently detailed information, the Okmeydanı residents increasingly relied on their own observations and investigations. For example, the absence of observed damage during the Kocaeli earthquake in August 1999, when "no single house, not even a garden fence had any single crack or damage", was interpreted as an indicator for a low seismic risk of the neighbourhood.

> *"So what we know from experience is that we are not a risk area. Our experience with past earthquakes proves us this. But you are all experts in that, please make your own investigations and tell us. We are very happy to cooperate with you." (Ali, Okmeydanı)*

10 Despite being highly suspicious of government actions, the Okmeydanı-based neighbourhood association expressed their goal of preparing and promoting an alternative urban planning that "incorporates the idea of risk mitigation", and that is more socially equitable, sustainable and based on close cooperation with independent geoscientists.

> *Researcher 8: What would change if we [as independent scientists] would say that this is indeed a high-risk area?*
> 15 *Ali: First of all we would thank you that we have the chance to finally learn about the threat. Then we would of course cooperate with you, and would like to hear from you what you would suggest. We would like to hear that, because of course for all of us human life is the most important thing. Please come to us with your suggestions and let's think together what can be done.*

20 Through conversations with local inhabitants, the ALErT geoscience researchers were exposed to a social framing of Istanbul's seismic-hazard preparedness dilemma that was very different from their own geological and geophysical perspective. Issues that emerged as alternative dimensions of the seismic-risk problem – and that were not visible to the researchers before the field encounters – included recognition of local concerns about the continuity of social networks, the importance of co-determination and transparency, and trust in authorities in charge of mitigation measures. Significantly, as 25 is demonstrated in the final exchange between researchers and the Okmeydanı neighbourhood representative, the direct involvement of geoscientists in addressing the 'seismic problem' was encouraged, alongside the desire among residents for "actionable" communication.

**5 The evaluation workshop: lessons from Istanbul's seismic suburbs**

Following the urban field visit, a round-table workshop allowed the geoscientists to reflect on the residents' issues and 30 explore through semi-structured group discussions their own contrasting perspectives on the motivations and responsibilities



of communicating their science to the public. Here, we report, through the direct comments of the ALErT researchers, four principal areas of concern that emerged in those group deliberations.

## 5.1 Impact of seismic risk communication on individual preparedness

The fact that inhabitants of high-risk areas often do not translate an increased risk awareness into adjustment behaviour was
mostly familiar to the workshop members. Yet, the various factors influencing how inhabitants ultimately act upon geoscientific information were much more apparent to the participants in the context of the field excursion and led to discussions about the basic nature of geoscience communication:

> *Researcher 8: If you would have asked me before the workshop, I would have said geocommunication is*
> *contributions, papers, conferences...But now it is gaining much more body with the public.*
> *Researcher 3: I am not even sure if geoscientists' answers are necessarily involved. I think that politicians' and the*
> *public's communication about geoscience issues is also geocommunication in a way.*

Turkish geoscientists within the group corroborated and substantiated resident's statements about the deficiencies in seismic
risk communication. According to them, media coverage on seismic risk often gives misleading, partly contradictory information, including a severe lack of "actionable" communication. Particularly information on prevention measures and geoscientific background information were described as not easily accessible and not sufficiently user-friendly for at-risk communities. Also, participants perceived a general weakening of prevention messages after a short "window of opportunity" following the Kocaeli and the Duzce earthquake. Aspects mentioned were a diminishing media coverage of the
topic, a focus on "disaster headlines" and a decreasing visibility of public education campaigns, such as "earthquake simulation buses" and the promotion of "family action plans".

> *Researcher 5: I remember that just after the big earthquake in Duzce they had films, advertisements. They had some*
> *commercials. Some information what we can put in our backpacks, how to make emergency plans (...) but now*
> *there is nothing. Everybody forgot about it.*
> *Researcher 2: After 16 years, of course everything changes.*

Several participants also criticized an insufficient practical application of public prevention measures from governmental institutions. The existence of "nice looking reports" from institutions such as AFAD were seen in clear contradiction to the
actual implementations on the ground.



> *Researcher 5: There is no application. They [the governmental authorities] say: "Yes, we have to do that."(...) Yes, good plan, good application. And when a natural hazard or an earthquake is coming, there is no application. It is written, but there is no application.*
>
> *Researcher 6: There is no continuity.*

While the group expressed their comprehension for inhabitants that have difficulties identifying with the form and content of seismic risk communication, they also criticised a lack of motivation to take individual adjustment measures. Aspects named were a reluctance to go online and actively search for prevention measures in their everyday life, but also a tendency to "listen and forget" about information or to rely on "fatalistic arguments". Supporting the published research literature, some

argued that these risk perceptions are also culture-specific, as emerges in this exchange with the facilitator:

> *Researcher 5: I do not think that ordinary people like Sedat will go to the Internet and type in "What can I do in the case of an earthquake? What is the emergency plan?" I don't think so. For Turkey it is a little bit hard to get the attention of the people about these serious things.*
>
> *Facilitator: Why? Is it a cultural thing in Turkey?*
>
> *Researcher 5: For example if you want to give them some important information, if you want to inform them, they easily forget about it. They don´t want to do any action about it. They just listen and then forget. There is no prevention, there is no application. So it is a little bit about the culture.*
>
> *Facilitator: Do the others accept that? Before we judge the culture in Turkey...is it maybe different to the US or*
> *Japan?*
>
> *Researcher 6: One aspect is that culture is also affected by religion. When you say "There will be an earthquake", they say: "Oh, if it is going to happen, it will come from God."It is a faith and they tend not to do anything to avoid the bad circumstances of these events.*

**5.2 The role and responsibility of geoscience-communicators**

Despite broad agreement within the group on the relevance and importance of reaching at-risk communities, there was an intense discussion about the appropriate way and level of engaging with the public. Much of the debate therefore centred on the participants' individual understandings of the role and responsibility of "geo-communicators", and what implication this has on their professional life.

> *Researcher 1: If you know that something will happen (...) that many people could die (...) you will have to communicate that. You have to communicate that in order to prepare people.*



Despite an awareness of the modern push for the democratization of knowledge, some participants found it crucial not to blur the borders between scientists and non-scientists and to retain their role as "objective experts".

> 5    *Researcher 11: (...) I think you should do your best to improve your analyses and get proper results and publish and explain these results to proper people. For example, the government or the administration. And these people should know what to do with this. You can give them suggestions what you think is the best idea to use the results and how to protect the people, but the decision belongs to them.*

10    Some of the group considered geocommunication as a rather "one-way", linear transfer of "geoscientific expert knowledge", restricting geocommunication to "the provision of correct data" and "recommendations" to decision makers (government, civic administration, selected media representatives) who then "should decide what to do with the information".

> *Researcher 4: In my humble opinion, science has something to do with knowledge,. Policies, hazard mitigation, those*
> 15    *are things related to judgement, to decision-making. Those are two completely different things.*

For these participants, a direct engagement with residents, particularly in politicized contexts, was considered as negatively affecting this role, and potentially risking a loss of reputation, trust and scientific credibility due to actual or perceived advocacy positions. Others, however, whilst acknowledging these fears, stressed instead the "moral and professional duty" to
20    directly provide their expertise to communities, especially in situations where inhabitants face an acute risk and openly request closer collaborations with scientists. For them, there was a "risk of losing public trust" when not reacting on shortcomings of communication, as this exchange reveals:

> *Researcher 8: A hypothetical case, let's imagine the scientific community has a very clear view that the Marmara*
> 25    *earthquake is going to happen in five years time, and it is going to be magnitude 8. Then what is your responsibility, when people are not reached by standard geoscience communication? This is how I face this problem. Then you really have two push the boundaries and tell the people that they should move away from the boundary (...) but I am already in the activist part.*
> *Researcher 2: You're looking at the human aspect, not at the scientific aspect. As a human being, when you see that*
> 30    *something bad will happen very soon, then of course you will push people and try to fix the problem, but this is the human aspect and not the scientist's aspect. As a scientist you just have to do the research, get the information and share it.*



> *Researcher 8: But I absolutely don´t feel like this – this is my scientific part and this is my human part (...) I don't understand why geoscience should be communicated in a very specific, narrow way, for example centred on geohazards. Then people might know something about the physics happening, but they don't really do anything in their daily lives. And this is the challenge.*

> *Researcher 7: You could make sure that you inform the public better, so that they can find a way around this corrupt system so that people are informed to really make decisions.*

> *Researcher 10: But this is really complicated.*

Many of the participants shared this view that despite a high responsibility of geoscientists to engage in a more "interactive process" and a "better communication", it was as a very challenging task. Participants expressed an insufficient knowledge about how to initiate such exchanges, how to methodologically approach them, and how to understand and approach the complexity of audiences and their cultural settings.

## 5.3 Lack of intermediaries and interdisciplinary collaborations

Despite the fact that the majority of the group agreed on geoscientists responsibility to interact more effectively with at-risk communities, the round-table discussions brought to the fore a concern among the participants of not having sufficient communication skills to successfully connect with lay audiences. Only a few participants could give firsthand examples of science-public interactions beyond casual conversations with friends and family members; some mentioned occasional encounters with local residents in the course of their field work, incidents in which they "had to get information from local people", and were asked to "explain" what "they are doing". Beyond these exchanges, interaction with different audiences were viewed as a "rather unknown territory". Major constraints named by the participants were an insufficient knowledge about "how to methodologically approach such exchanges", "which media and format to use", "how to understand and how to respond to the complexity of audiences and their cultural settings", or simply to be locked in the "geoscience world" - a metaphor for a rather "closed system". Debates emerged about whether to "pinpoint the communication talents" within the geoscience community or to engage in interdisciplinary research collaborations. Yet, such interdisciplinary collaboration was described in the discussions as being highly relevant in order t It was suggested to broaden collaboration networks for example with social scientists, but also with media representatives, artists or NGOs, who were seen as promising "intermediaries" or "translators" to more effectively share knowledge with people "on the ground".

Despite many participants' wishing to acquire skills for a more interactive engagement with local communities, the barrier was often cited.



*Researcher 10: Our responsibility is to produce science and use other scientists who can talk to people, like anthropologists, sociologists or people who have studied philosophy, psychology, this kind of stuff... My point is that we need a bridge to communicate with the people. We cannot communicate directly. We need a translator.*

*Researcher 3: Or translate it ourselves.*

*Researcher 2: It won´t be that easy for us.*

The proposal "to use" external interlocutors to help facilitate geoscience communication was countered by some individuals, worried that working with other groups might negatively affect the quality of messages. For example, collaborations with journalists "to reach people", were deemed important, albeit limited by the constraints of media agenda-setting and "loss of

information" from the perceived insufficient "accuracy" of journalistic writing. This scepticism towards the scientific quality of journalistic writing was also assigned to social media representations. Despite this, group members accepted that only a small minority of people read scientific journals or news reports from research institutes – outlets to which the participants assigned the greatest trust in terms of 'properly' conveying scientific messages.

*Researcher 6: The translator should be capable enough to translate the geoscientific language correctly. So maybe it can be better as geoscientist if we could be able to directly tell the way it is to the public rather than using an agent in between.*

*Researcher 10: If we want to bring the education to the people possibly the best point to start is to use the universities which have a facebook page. And specifically when talking about earthquakes they should choose simple words.*

*Researcher 6: I am not using Facebook. That kind of network wouldn't be able to reach people like me. Ok, we are a minority for sure. If I am going to follow some news about earthquakes, this wouldn't be Youtube. I have never followed news about earthquakes on Youtube. I prefer an earthquake observation centre. But I am educated on this and my options are different.*

*Researcher 8: But what if you were Sedat?*

*Researcher 6: If I would be Sedat I would prefer... daily newspapers.*

*Researcher 8: All types of media! You have to reach the TV, the newspaper, facebook, twitter...everything. That is difficult.*

In addition to discussing the multiplicity of media communication networks, debate over the need to collaborate to achieve

more effective geoscientific outreach led to strong exchanges within the group, with some of the participants finding it unsatisfactory to depend on "agents" to share their knowledge with the public. Instead, some argued for a better appreciation





of knowledge exchange and mutual learning within networks, and for more awareness of the ethical aspects of communicating science, with some describing it as "something you just do naturally", which is "not optional".

> 5
> *Researcher 6: Why do you think that only the geoscientists give the information? Maybe there are things that you don't know, and that only an ordinary person knows.(...) For example when you go to the field, (...) to a little village, if you are working on a recent event of that region, you go to the manager of the village, and you talk to him, for example "Have you ever had any floods in this area?" It is a communication situation and you learn from a person that is not a geoscientist.*
>
> *Researcher 11: If you find a way how to communicate with the people, the agent is not always necessary.*
>
> 10
> *Researcher 1: It makes much more sense to bring people into the topic. The problem is not that they don't know that an earthquake might happen. That is not the problem. The problem is that they have to deal with that problem. And usually the best way to motivate people is a playful way. That the people can be active together and learn at the same time. It is like language learning. If you don't apply it together with others, then you forget it.*

## 5.4 Constraints and the appreciation for transdisciplinary approaches

A final strong sentiment that emerged from the workshop discussions was the expectation among the group that geoscientists ought to engage in communication and outreach activities together with other disciplines and with at-risk communities. Yet, all participants expressed their concern that the practical side of such inter- and transdisciplinary activities wasn't sufficiently supported by institutional frameworks of universities or research institutes. Participants mentioned multiple hurdles, commonplace in science communication surveys: "Maintaining a career", "time pressure", "specialization", "publications

mostly for academic journals". Despite considering that communication with different publics was a "moral obligation", not least because scientists are mostly "being financed by taxpayers", some participants underlined a perceived lack of "reward" for such engagement. Communication, in this context, was still viewed as something "extra" or "private" within the academic science landscape. For example, writing about geohazards and risks using social media channels was perceived as something associated with "leisure" or "sacrificing leisure time"; answering scientific questions within social networks

something that "you simply do" because of social expectations.

> *Researcher 8: (...) It is our responsibility. But the problem is: We are not paid for that. We have to maintain a career as well. And this is only one of the little aspects that are very relevant. We have to do it for the sake of it. We do a lot of things for science which are for free. And we also have a hard time to maintain a pace...and to do*
>
> 30
> *publications, to find the next position and so on. So it is a very difficult balance.*
>
> *Researcher 7: There is no real reward.*





*Researcher 8: It depends on how you interpret reward.*

It was also described that outreach training usually focuses on the development of presentation skills, of raising an awareness for the necessity of a more user-friendly language, the importance of storytelling or the visualization of research findings. These skills were regarded as fundamental and important, but they were considered insufficient when it came to the challenge of meaningfully connecting with and learning from widely different audiences. One participant mentioned that "ideally we should have 48 hours a day", "to educate in schools, to educate the media, to educate the politicians" and "to learn what is relevant for them". Encounters that provide opportunities for mutual learning, whether by involving local practitioners or other disciplines within university frameworks, were seen as an uncommon praxis. For the ALErT researchers, the idea of a transdisciplinary communication training framework represented a distant "ideal", and in that regard, even the workshop itself was considered an "unfamiliar event".

## 6 Discussion: emerging principles and practices

There was general agreement among the participants of the relevance of more effectively reaching out to at-risk communities, and also an acknowledgement of the difficulty in deriving a standard formula for geocommunication training. Several underlined the view that "every scientist has a different level of capacity or ability in order to reach the public", and the notion of a strict allocation of time spent on communication activities was seen as unrealistic. Yet, all participants expressed their wish for a serious reappraisal of some core principles:

### 6.1 Principle 1: a more holistic perspective on geocommunication

Geoscientists need to evolve a greater awareness for the various audiences and the messages that confront them. The generation and provision of technical expertise remains the primary responsibility of geoscientists, but more integrated approaches are needed to incorporate the social and cultural dimensions of risk. Such a holistic approach to the idea of "interacting with the public" is deemed to include a greater sensitivity towards the factors inhibiting successful science-public-interaction: Insufficient knowledge about the target groups of communication, weak empathy for citizens concerns, or a strong politicization of a topic can lessen the impact of risk communication on scientific literacy or individual behavioural change. In particular, integrating local perspectives into communication strategies was considered as essential in order to better frame messages and ultimately reach wider audiences. Given the interdependences of hazards and societal concerns, geoscientists – as specialists in complex interacting systems – were seen as being entirely capable of raising people's interest for these challenges of the dynamics of the Earth and of the places where people live.

### 6.2 Principle 2: the need for inter- and transdisciplinary collaboration



Despite the expressed primacy of geoscientists in the geoscience communication process, collaborations that forge links with citizens, scientists from different disciplines, planners, politicians etc., and which help to exchange views and address problems collectively, hold the potential to "unlock" the societal impact of (geo)scientific expertise. Yet, according to the workshop participants, "more practical experiences" with these alternative formats are needed. Two approaches were deemed

important.

Firstly, many participants encouraged to actively integrate lay people into the development of communication strategies. In the context of the field excursion to Zeytinburnu and Okmeydani, local citizens made clear that excluding local groups from technical assessments and participation can create resentments, limit valuable information sources, and undermine the legitimacy and outcomes of communication and mitigation efforts. Paying more attention to these important, but often

invisible concerns relevant for inhabitants and integrating their feedback (e.g. whose interests are served or threatened by various natural hazards, and to what degree information have an applicable utility for inhabitants) can avoid stereotypical depictions of target groups or even worse "the public". Similarly,  collaborations with artists, journalists and NGO´s were described as promising, as such groups often have a longer and more deep-seated experience or skills in approaching various publics and as they can bring important insights and perspectives into the design of risk communication.

Secondly, the role of interdisciplinary linkages was perceived as significant. In the context of collaborative projects, synergies can be elaborated, for example by critically reflecting methods, terminologies and concepts that are often not questioned within mono-disciplinary frameworks. Specifically highlighted was the integration of insights from empirical social sciences that can help to give geocommunication strategies a more theoretical foundation to effectively target audiences, develop clearer messages, and adopt the most appropriate channels and platforms of communication.

**6.3 Principle 3: a stronger institutionalization and assessment of inter- and transdisciplinary communication approaches**

Despite the fact that "interdisciplinary" and "transdisciplinary" collaborations are terms frequently used in public outreach descriptions, there was broad agreement among the ALErT researchers that in reality, critical practice and evaluation of such collaborations are often weak. A view that emerged from the workshop was that the university should be a "central player" in

novel science communication approaches. Three major aspects were particularly relevant for the participants:

Firstly, the communication workshop clearly showed that network-based processes can be challenging, undirected, time- and resource-consuming. In order to facilitate inter- and transdisciplinary learning, geoscience communication training was seen as requiring curricular changes. Such training ought not only to inculcate skills for an effective transfer of knowledge (presentation-, storytelling- or visualization techniques), but also develop researchers' ability for "story-listening" and their

sensitivity for the complex societal dimensions of communication. Integrating relevant actors from outside was seen as



equally important as a regular training in using social media such as blogs, social networks such as facebook and twitter for actual interaction with lay people.

Secondly, there was broad consensus among participants that joint research projects within a university framework could be ideal places for communication training. These research projects should not only bring together scientists of different disciplines, but also create forums for deliberative approaches that link scientific expertise with community concerns. Some group members argued that universities should "have the courage" to provide knowledge dedicated to communities at-risk in novel forms, for example "by more frequently using their web- and Facebook pages".

Thirdly, novel methodological approaches in science-public-collaborations have to be developed and need an in-depth evaluation. One participant critically noted that, while many outreach activities are being done and labeled as innovative, no one knows "if they really create an impact."

## 7 Conclusion

Conventional seismic risk communication tends to be focused on the conveyance of "geo-facts" (Stewart and Nield, 2013), but, as the ALErT geo-communication workshop demonstrates, geoscientific "matters of fact" are generally strongly intertwined with societal "matters of concern". This results in knowledge configurations that strongly influence the efficiency of geoscientific information transfer. A way to counter this tendency is to integrate local perspectives into the design of communication approaches. Through critical conversations with urban residents, the participants of the ALErT workshop recognized important community-centred issues and concerns that had previously lain outside their geoscientific perspective. Maintaining the value and integrity of the disciplinary knowledge (sound science) whilst at the same time adjusting to specific socio-cultural contexts requires a transdisciplinary mode of communication training. Field-provoked communication workshops, community-centred participatory knowledge exchanges and self-reflection on disciplinary practices and paradigms are elements that can readily be incorporated into geoscience communication training more widely. By fostering a more nuanced understanding and awareness of the complexities of science-public collaborations, the next generation of geoscientists can start to develop more fruitful ways to build genuine resilience among those most at risk.

*Acknowledgements.*

*The present study and the associated doctoral research (JI) were generously funded as part of the Marie Curie Integrated Training Network on 'Anatolian pLateau climatE and Tectonic hazards' (ALErT). The authors express their gratitude to the ALErT doctoral and postdoctoral students for their active participation in the workshop, and especially to Dr. David Fernández-Blanco for comments on the manuscript. The authors are indebted for the advisory contribution from Dr. Martha Blassnigg, whose untimely death during the study was a major loss for us, and for the wider transdisciplinary science community.*



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
