# Peer review of "From geoscientific 'matters of fact' to societal 'matters of concern': a transdisciplinary training approach to communicating earthquake risk in Istanbul (Turkey)"

_Natural Hazards and Earth System Sciences, 2016_

## Referee Comment (RC1) · Anonymous Referee #1 · 29 Jan 2016

As a scientist who does science communication but is not trained or educated as a science communicator, I basically enjoyed reading the manuscript, which is well structured, well written, self-contained, and self-consistent. The ALErT program must be a nice opportunity for young earthquake scientists to learn how to communicate with lay people with various background. In addition, I advocate recommendations given by the authors in Section 6 (Discussion); the authors raised important points for an effective science communication on earthquakes or probably in general. With this, I have only few minor comments to be addressed before the eventual publication. Please find my comments as enumerated below.
[Figure]

1. My only major concern is that the format of this manuscript does not look like a scientific paper I usually read, which are physics-based papers, but like an opinion article. For example, number of scientists participated in the project is 12, too few to do any statistics.

2. The ALErT program itself is nice and what ALErT program did is nicely described. However, motivations for this project is not described. I would like to know, for example, 1) whether there are previous projects similar to ALErT, and 2) why ALErT targets young scientists.

3. The authors nicely summarized the outcome of the ALErT project, I am not sure whether the output of the project can be generalized to a communication with people in other at-risk areas such as San Francisco, Los Angeles, or Tokyo. If there are similar previous studies, the authors may want to compare the result of the present study with previous studies.

4. Page 4 line 23 proscriptive: prescriptive?

5. Page 12 line 29 AFAD: What does AFAD stand for?

6. Page 15 line 25 in order t It was suggested...: Need a rerun of a sentence including this.

---

## Referee Comment (RC2) · Anonymous Referee #2 · 24 Feb 2016

Being a geoscientist also trained in risk communication, I am very happy to see such type of paper. I would like to thank you for this very interesting piece of work. It highlights the very important topic of geoscientists as communicators. However, I believe that some improvements (transparency of the methodology, inclusion of relevant literature, deeper discussions) are needed to make this paper excellent. Please see my specific comments bellow.

The title is somehow misleading. I think it is very well phrased and striking but not very appropriate in my opinion. Your discussion points are not only valid for young geoscientists' training but also for the general conduct of geocommunication. Moreover, although you advocate for a transdisciplinary training approach in your discussion, you use an example (the ALErT project) that could be perceived as not really transdisciplinary as most participants are geoscientists. With this title, someone can expect to be introduced to a successful example of social sciences/geosciences training approach.

First paragraph of the introduction: all these sentences deserve to be supported by literature. These are bold statements that can be argued. If those are not taken from literature, please clarify that they reflect your point of view or your experience. This comment is also valid for other sentences throughout the paper, for example page 3 lines 3-6.

Also related to literature review, I think you do not put your work in perspective of the huge discussion that is already existing on the role of natural hazard geoscientists as communicators and on the need of transdisciplinary approaches. I miss a literature section on this. A lot of scientists and organizations have already advocated for these needs. As an example, the paper of David Liverman ("Communicating Geological Hazards: Educating, Training and Assisting Geoscientists in Communication Skills") or the "Geoscience and Natural Hazards Policy" Position Statement of the Geological Society of America. Moreover, related to the L'Aquila earthquake, the debate was fierce. And very apropos, you have already the sentence to introduce such literature review section (page 6, line 11).

Page 4: small inconsistency between line 19 and line 31. You say that around half of the researchers come from Turkey but then you state that most of the participants are unfamiliar with the Turkish cultural context. The use of "most" is probably not appropriate.

I miss a methodology section about the interviews with the inhabitants. This is a question of transparency. You should state how much people were interviewed, who they were, by whom they were interviewed, what were the questions. In more general terms,

what was the research design? In this section, the research design of the workshop should also appear, e.g. mention of focus group method, the professional identity of the facilitator.

I like the fact that some of the interviewees' statements appear in the bulk of the paper itself. However, here again I see a transparency issue. The reader cannot judge if the chosen pieces are representative of the whole sample. For sure you mention additional testimonies to make your point: for example, page 9, lines 25-30; page 12, lines 14-21; page 15, lines 14-25; page 16, lines 7-13; or page 17, lines 19-25. I would like to see those statements as well. Maybe in a supplementary material file that could be accessed in the online version of the paper if you think that a table directly in the paper is too much.

Also related to the interviewees' statements, I found that they are not always very well chosen. For example, page 8 in relation to participation. The statement that is summarized in the 26 seems to be a better highlight of the complaints than the one provided in line 23-24.

Page 12, line 2: "four principal areas of concerns", what were the others ones? Explicitly mention why you chose to present only those.

Page 12, line 18: here by "participants" do you refer to only the Turkish ones or the whole group? I would also have liked to know which of the researchers are Turkish or not. As you mention, the cultural aspect is important, so it would be interesting to see if there are perception's differences between the participants in relation to this point and also to see some discussion about that.

On the same line, I would have expected a deeper discussion section as well as greater use of the literature on some of the issues brought up by the participants, e.g. cultural aspects (page 13, lines 16-23) or the role of the geoscientists in communication (pages 14, lines 31-32). Additionally, you do not challenge some points: e.g. "strict allocation of time spent on communication activities was seen as unrealistic". You certainly know

that outreach activities are compulsory for some Horizon2020 projects and thus time must be allocated to those.

Page 18, lines 23-25: provide references.

Page 19, section 6.3: Examples of universities proposing a joint technical and science communication master could be added as a good practice.

Technical corrections: Page 6, line 22: one "for" too much. Page 15, line 25: "in order t It was"

---

## Short Comment (SC1) · 21 Mar 2016

Interactive comment on "From geoscientific "matters of fact" to societal "matters of concern": a transdisciplinary training approach to communicating earthquake risk in Istanbul (Turkey)" by J. Ickert and I. S. Stewart

J.Ickert and I.S. Stewart We would like to express our gratitude to the two reviewers for their detailed and constructive feedback. Both comments were highly valuable for a critical reexamination of major aspects of the manuscript. Especially the recommendation to further elucidate the methodological approach and our epistemological

understanding of transdisciplinary research, as well as the recommendation to deepen our literature review for certain aspects of the paper turned out to be very helpful and in our opinion helped to contribute to the relevance and actuality of this paper.

Our replies to the reviewers comments are provided in italics below.

RC1 anonymous As a scientist who does science communication but is not trained or educated as a science communicator, I basically enjoyed reading the manuscript, which is well structured, well written, self-contained, and self-consistent. The ALErT program must be a nice opportunity for young earthquake scientists to learn how to communicate with lay people with various background. In addition, I advocate recommendations given by the authors in Section 6 (Discussion); the authors raised important points for an effective science communication on earthquakes or probably in general. With this, I have only few minor comments to be addressed before the eventual publication.

We thank the reviewer for this positive feedback.

Please find my comments as enumerated below. My only major concern is that the format of this manuscript does not look like a scientific paper I usually read, which are physics-based papers, but like an opinion article. For example, number of scientists participated in the project is 12, too few to do any statistics.

We agree that this research paper may be unfamiliar for NHESS readers. Therefore, we would like to highlight again the aims of this project, as they motivated the non-traditional approach we have taken. It is widely discussed in social science literature on effective risk communcation that a top-down knowledge transfer of scientific information to at-risk communities is having little or no influence on their risk adjustment behaviours (Moser, 2014; Solberg et al., 2013; Wood, 2014), and there is broad agreement among numerous well-recognized risk communication researchers, that novel approaches have to encompass participative methods to ensure the effectiveness of risk communication (Wachinger et al., 2013). Helga Nowotny, one of the spearheading figures of the discourse on transdisciplinarity claims for a more "socially robust" science communication (Nowotny, 2010). Although the use of transdisciplinary research methods/multistakeholder approaches are widely promoted, there are considerable shortcomings on knowledge about their actual application within the the field of risk communication (Werlen, 2015; Arvai, 2014). Therefore, case studies and practical experiences in different cultural contexts are highly needed in order to find out under which conditions these appraoches can be applied and in order to develop valid methodologies for organizing transdisciplinary approaches. This paper is an attempt to provide a practice-based reflection on how young geoscientists think about and experience transdisciplinary approaches in "real world-settings". It is important to underline that it was not our intention to follow a quantitative approach. Our approach was to apply qualitative methods in a small group of workshop-participants that facilitate the articulation of opinions that are often overlooked. In order to further elucidate this approach, we found it necessary to more clearly explain the workshop methodology in the beginning of section 4 of our paper:

In order to gain a more nuanced understanding of the sociocultural dimension of risk communciacation in Istanbul, we decided to apply a combination of two qualitative social science research methods in the context of the workshop. In order to enhance the group participants understanding of local perspectives that are generally not integrated into seismic risk communication, they were asked to do field-based, semi-structured interviews with residents of the Urban Renewal neighbourhoods. In addition, to support a process of knowledge co-creation, we organized a series of moderated focus group discussions to more openly voice different topics and concerns related to seismic risk and its mitigation. In order to document the process for further analysis, both the field visits as well as the focus group discussions were recorded on camera by the assistant moderator.

Field-based narrative interviews with local stakeholders In late May 2015, under the guidance of local urban historian Orhan Esen, the 14 geoscience researchers under-

took a half-day field visit to the gecekondu districts of Zeytinburnu and OkmeydanÄś. At important spots of the visited neighbourhoods, Orhan Esen gave short interactive lectures about the historical context of informal settlements and on the way seismic risk mitigation measures are being locally implemented. In addition to this valuable local information, the visit gave the participants a first-hand picture of the building stock of both neighbourhoods and provided the opportunity to meet several inhabitants and community representatives. During the first stop in Zeytinburnu, the participants initiated two extended interviews with inhabitants of the Urban Transformation area "Sumer Mahallesi". The interview partners also guided the group to the old gecekondu part of the neighbourhood, that was not yet transformed. Both interviews were translated from Turkish into English by Orhan Esen. The second field stop was the neighbourhood of Okmeydani. In a 2-hour roundtable set-up, the participants had the opportunity to interview Ali Çetkin, chairman of the OkmeydanÄś-based neighbourhood association "OkmeydanÄś Çevre Koruma ve Güzelleştirme Derneħi", on his and the associations perception of seismic risk mitigation, given their specific locale. In addition to his detailed statements, he provided a broad array of visual materials, such as maps, newspaper articles and public announcements. The emerging discussion was moderated by Orhan Esen, who also translated the Turkish contributions into English. During the stay in both neighbourhoods, participants were asked to take detailed field notes.

Focus Groups On the back of the field visit designed to take the geoscience researchers to the edge of their academic 'comfort zone', the authors facilitated two 90-minutes focus group sessions to explore the perceptions and attitudes of the ALErT investigators to the prospect of communicating their science more broadly. The initial focus group discussion took place on the afternoon of the field visit and and aimed to provide the participants a framework to reflect the field experiences and to voice their own individual views and concerns about their potential roles and responsibilities as geo-communicators. There then followed a 5-day technical field course along the North Anatolian Fault during which the participants were encouraged to have informal discussions among themselves about the broader issues of geo-communication. At the

end of the field school, a second focus group was organised to elucidate the groups reflections on effective approaches of communicating hazard science to at-risk communities. Both groups were structured around a set of preconceptualized questions, but the discussion itself was free-flowing. Both focus groups were moderated by the workshop facilitator Prof. Iain Stewart, who ensured that the variety of different ideas and opinions from as many group participants as possible could be voiced. He also facilitated an inventory of the discussion positions of the group, which were later structured on a flip-chart. The following paragraphs of section 4 will outline the key aspects that emerged through the interviews and group discussions among workshop participants and inhabitants. These summarize local perspectives on the mitigation process relevant to consider when developing communication strategies for (and with) at-risk communities. Section 5 will discuss the topics and concerns that emerged during the second focus group session and summarize the recommendations elaborated by the participants.

The ALErT program itself is nice and what ALErT program did is nicely described. However, motivations for this project is not described. I would like to know, for example, 1) whether there are previous projects similar to ALErT, and 2) why ALErT targets young scientists.

Thank you for these important questions. We have added the following text in order to further elucidate the motivations of the ALErT project and why it targets young scientists.

The geoscientists in question were a group of doctoral and postdoctoral researchers engaged in a Marie Curie Integrated Training Network on 'Anatolian pLateau climatE and Tectonic hazards' (ALErT). This network aims to provide state-of the art research results on the complex interaction between tectonic and climatic processes which influence the morphologic evolution of the Central Anatolian Plateau (CAP) in Turkey and associated natural hazards. ALErT's emphasis on natural hazards – principally earthquakes, landslides and flooding – means that in addition to receiving training in

advanced methods of geoscience data acquisition and field investigation, the young geoscientists are expected to develop expertise in effective science communication, particularly within transdisciplinary frameworks. The severe consequences of the 2011 Tokoku earthquake or the L'Aquila earthquake highlighted how crucial it is to critically reflect paradigms of risk communication and to equip scientists with the competence to communicate with non-specialists in ways that are not only scientifally sound, but also socially robust. ALErTs underlying assumption is, that particularly young scientists need to be trained in using innovative methods that help to better respond to the pressing demands of an increasingly vulnerable world.

The authors nicely summarized the outcome of the ALErT project, I am not sure whether the output of the project can be generalized to a communication with people in other at-risk areas such as San Francisco, Los Angeles, or Tokyo. If there are similar previous studies, the authors may want to compare the result of the present study with previous studies.

It is important to outline that this paper did not outline the outcome of the ALErT project, but of a communication workshop that took place in the context of the ALErT project. Regarding the generalization of the outcomes of the workshop, we would like to highlight the following points.

As seismic risk communication in Istanbul is highly politicicized, mainly due to the cities contested Urban Renewal projects and a lack of citizen participation, we do not think that our findings on the specific risk perceptions of inhabitants of Urban Renewal areas can be directly applied to other at-risk areas. Yet, many factors that we discussed and that impact risk adjustment behaviour, such as the lack of trust, a lack of "actionable" approaches, a lack of transparency and established forums for science-public interaction or media influences, corroborate the findings of other social science studies on sociocultural factors influencing risk perception and risk behaviour, such as Green, 2008, who examines risk attitudes of Istanbuls residents of informal settlements, or Wood, 2014, who introduces a broad array of factors that motivate risk adjustment of
US-citizens, or Lin et al., 2015, that provide an interdisciplinary study on the social and physical determinants of seismic risk in Taiwan. Another example we highlighted was the comparative study on social representations of earthquakes in Izmir, Seattle and Ozaka by Joffe et al.., 2013. Here, the authors show that despite broad public awareness of an increased earthquake risk, Turkish residents are far less likely to undertake anti-seismic preparedness or mitigation measures than their counterparts in California and Japan, outlining that risk perceptions patterns cannot be easily generalized, but are influenced by numerous variables that differ according to cultural context, sociodemographic situations, psychological factors, experiences with past earthquakes etc. We hope that our paper pointed out how important it is to put these different insights into perspective, in order to gain a more nuanced understanding of science commmunicators for the multiple sociocultural vectors shaping risk adjustment behaviour.

Page 4 line 23 proscriptive: prescriptive?

This was changed as per the reviewers suggestion.

Page 12 line 29 AFAD: What does AFAD stand for?

AFAD is theÂăDisaster and Emergency Management Presidency of Turkey (Turkish: Afet ve Acil Durum Yönetimi BaşkanlĜħĜ). The text now reads as follows: "The existence of "nice looking reports" from institutions such as the Turkish Disaster and Emergency Management Authority (AFAD) were seen in clear contradiction to the actual implementations on the ground."

Page 15 line 25 in order t It was suggested...: Need a rerun of a sentence including this.

This was changed as per the reviewers suggestion.

RC 2 anonymous Being a geoscientist also trained in risk communication, I am very happy to see such type of paper. I would like to thank you for this very interesting piece of work. It highlights the very important topic of geoscientists as communicators. However, I believe that some improvements (transparency of the methodology, inclusion of relevant literature, deeper discussions) are needed to make this paper excellent. Please see my specific comments below.

We thank the reviewer for these kind words.

The title is somehow misleading. I think it is very well phrased and striking but not very appropriate in my opinion. Your discussion points are not only valid for young geoscientists' training but also for the general conduct of geocommunication. Moreover, although you advocate for a transdisciplinary training approach in your discussion, you use an example (the ALErT project) that could be perceived as not really transdisciplinary as most participants are geoscientists. With this title, someone can expect to be introduced to a successful example of social sciences/geosciences training approach.

Thank you for this indeed very valid point. Although the workshop - conceptionalized by both a geoscientist and a cultural anthropologist – engaged several actors such as a local urban historian, earth scientists and local inhabitants in exploring different framings of seismic risk and its communication, it was not transdisciplinary in a strict sense. It rather provided a framework for the workshop participants to engage in a a field-based reflection on theoretical and practical implications of transdisciplinary approaches in seismic risk communication. We have integrated this important differentiation into the paper and changed the title to

"On the necessity of transdisciplinary approaches in seismic risk communication: Insights from a workshop in Istanbul′s Urban Renewal Areas"

Your feedback also motivated us to further highlight in the paper our understanding of transdisciplinary approaches in risk communication. There is a multitude of different definitions of the concept of transdisciplinarity, and numerous case studies with scattered insights. This and the use of transdisciplinarity as a "buzz-word" in research proposals are leading to a scientific devaluation of its idea. In order to confront this, we have included a section in which we define how we understand transdisciplinarity and

outline the relevance for a clear methodological framework to assess transdisciplinary projects.

The section now reads as follows: Our methodological attempt is embedded in a broader discussion on the role of transdisciplinary approaches in risk communication, that emphasizes moving beyond traditional hazard and risk management to involving communities and decision-makers in "extensive campaigns of knowledge exchange and communication" that will lead to practical solutions for communities (Lindenfeld et al., 2014). According to Bunders et al., 2010, shared set of principles of transdisciplinary approaches are a) Joint process initiated outside academia (government, industry, public, NGOs), or by scientists on an ill-defined, societally relevant, real-world problem, b) Joint problem definition (including definition of system boundaries), c) a method-based analysis of the complexity of a system (actor analysis, causal analysis, system analysis), d) Mutual learning enhanced in focus groups, round tables, expert sessions, stakeholder dialogues, etc.), and e) The construction of 'robust orientations' for the development of outcomes (Bunders et al. 2010 in Weber 2015). Research on effective risk communication has increasingly paid attention to the role of these transdisciplinary approaches (Weber, 2015; Arvai and Rivers, 2014; Kasperson, 2014; Dietz, 2013; Popa et al., 2014, Hagemeier-Klose et al. 2014). Also new engagement and transparency policies in the US and the EU, as applied in the Horizon 2020 Framework Programme for Research and Innovation, indicate this shift towards more pluralistic set of participants and upgrades in deliberation processes (Bostrom, 2014). This way of integrative risk communication approaches, so Arvai "shares much more in common with the kind of analytic–deliberative process outlined by the United States National Research Council (National Research Council 1996). Here, as much as deliberation is meant to improve the capabilities of non-experts, it is also intended to provide much-needed insight to risk assessments and their subsequent application to risk management." (Arvai, 2014). Yet, these processes are challenging, time consuming and require innovative strategies to ensure valid and reproducible results (Weber, 2015). In the the case of Istanbul, such transdisciplinary approaches in risk communication

seem to be still far from reality, as indicated in the next section.

First paragraph of the introduction: all these sentences deserve to be supported by literature. These are bold statements that can be argued. If those are not taken from literature, please clarify that they reflect your point of view or your experience. This comment is also valid for other sentences throughout the paper, for example page 3 lines 3-6.

Thank you for this important feedback. We have supported our findings with references and included a more in-depth literature review into the introductory section, which now reads as follows:

As reflected in the concept of "shifting baselines", an important paradox of hazard communication is that the more effectively a potential physical threat is made public by the scientist, the more readily the scientific message becomes normalised into the complex, chaotic and contested discourses of daily life (Rost, 2014). By the same token, Wachinger et al. describe the severe consequences of a "risk perception paradox": It is widely assumed by risk communicators that a heightened risk perception or problem awareness leads to personal preparedness and consequently to risk mitigation behaviour. Yet, the relationship between risk perception and preparedness for actions is far more complex, indicating that factors such as the experience of a natural hazard or trust in authorities and experts are heavily shaping individual risk perception in often complex causal arrangements with many intervening factors (Wachinger et al., 2013). Although the standard communication approach of the so-called "deficit-model" (Frewer, 2004) in untouched by these real-world complexities, the majority of risk communication is still taking place in the form of a one-way transmission of risk information from experts to lay people that is likely to be ineffectual (Arvai and Rivers, 2014).

And: The contested nature of Istanbul's seismic preparedness exemplifies a general view emerging from disaster risk reduction research. This view stresses the necessity for hazard practitioners to pay more attention to the social and cultural dimension of

risk, and to analyse how, and if, adaptation and mitigation measures integrate local concerns (Moser, 2014; Krüger et al., 2015; Voss et al., 2014). Yet, despite various examples for a recent change of scientific and goverment risk communication practices towards an embeddedness of these practices within inter- and transdisciplinary frameworks (Bostrom, 2014), for the geoscientist charged with a responsibility to communicate the earthquake hazard, addressing the social and cultural dimensions of seismic risk is problematic (Werlen, 2015). Most hazard scientists are trained in the physics of natural processes and practised in intricate risk assessment procedures, but not in the nuances of political science or cultural theory, nor the sociology and psychology of human relations. For that reason, most geoscientists would regard it as beyond their realm and remit to confront the messy reality of how natural threats are translated and perceived by an at-risk community (The Royal Society, 2006; Jensen et al., 2008; Bentley & Kyvik, 2011; De Rond & Miller, 2005).

The difficulty is however, that for more than a decade social science studies indicate that there is little or no correlation between the provision of scientific information about risks and the adaptive changes in individual or community behaviour that would reduce risk (Slovic, 2000; Kasperson, 2014; Palm & Hodgson, 1992; Solberg et al., 2010; Wachinger et al., 2013; Fischhoff, 2012; Lichtenstein and Slovic, 2006). In other words, as is evident in Istanbul, a high risk awareness does not necessarily motivate personal or collective preparedness (Joffe et al., 2013; Green, 2008). The key message from social science is that simply getting the scientific message across to a vulnerable public is not enough – it needs new forms of public participation and of collaborative knowledge production (Weichselgartner & Kasperson, 2010; Arvai and Rivers, 2014). If geoscientists are to truly help an at-risk population adopt meaningful measures to protect itself, then arguably they need to design new strategies for public communication and community engagement (Wood, 2014).

Also related to literature review, I think you do not put your work in perspective of the huge discussion that is already existing on the role of natural hazard geoscientists as

communicators and on the need of transdisciplinary approaches. I miss a literature section on this. A lot of scientists and organizations have already advocated for these needs. As an example, the paper of David Liverman ("Communicating Geological Hazards: Educating, Training and Assisting Geoscientists in Communication Skills") or the "Geoscience and Natural Hazards Policy" Position Statement of the Geological Society of America. Moreover, related to the L'Aquila earthquake, the debate was fierce. And very apropos, you have already the sentence to introduce such literature review section (page 6, line 11).

This is a good and valid point. We have added a literature section that can be find under our answer to the reviewers recommendation for a revision of the title.

Page 4: small inconsistency between line 19 and line 31. You say that around half of the researchers come from Turkey but then you state that most of the participants are unfamiliar with the Turkish cultural context. The use of "most" is probably not appropriate.

We have changed this according to the suggestion of the reviewer.

I miss a methodology section about the interviews with the inhabitants. This is a question of transparency. You should state how much people were interviewed, who they were, by whom they were interviewed, what were the questions. In more general terms, what was the research design? In this section, the research design of the workshop should also appear, e.g. mention of focus group method, the professional identity of the facilitator.

We have added a detailed section on the methods applied in the workshop. It now reads as follows:

4 The workshop methodology: Gaining insights from Istanbul's seismic suburbs

In order to gain a more nuanced understanding of the sociocultural dimension of risk communciacation in Istanbul, we decided to apply a combination of two qualitative

social science research methods within the context of the workshop. In order to enhance the group participants understanding of local perspectives, that are generally not integrated into seismic risk communication, they were asked to do field-based, semi-structured interviews with residents but also to engage in informal conversations. In addition, to support a process of knowledge co-creation, we organized a series of moderated focus group discussions to more openly voice different topics and concerns related to seismic risk and its communication. In order to document the process for further analysis, both the field visits as well as the focus group discussions were recorded on camera by the assistant moderator.

Field-based narrative interviews with local stakeholders In late May 2015, under the guidance of local urban historian Orhan Esen, the 14 geoscience researchers undertook a half-day field visit to the gecekondu districts of Zeytinburnu and OkmeydanÄś. Important local information was gained through short interactive lectures by Orhan Esen at different spots of the visited neighbourhoods, that were followed by lively discussions. The visit gave the participants a first-hand picture of the building stock of both neighbourhoods and provided the opportunity to meet several inhabitants and community representatives. During the first stop in Zeytinburnu, the participants initiated two extended interviews with inhabitants of the Urban Transformation area "Sumer Mahallesi", that also directed the group to the old gecekondu part of the neighbourhood, that was not yet transformed. Both interviews were translated from Turkish into English by Orhan Esen. The second field stop was the neighbourhood of Okmeydani. In a 2-hour roundtable session, the participants had the opportunity to talk to Ali Çetkin, chairman of the OkmeydanÄś-based neighbourhood association "OkmeydanÄś Çevre Koruma ve Güzelleştirme Derneħi", about his and the associations views on seismic risk mitigation and its communication, given their specific locale. A broad array of visual materials, such as maps, newspaper articles and public announcements stimulated an extended roundtable conversation, in which the workshop participants raised prepared interview questions, but also more interactively engage in a discussion, that was facilitated through Orhan Esen and Prof. Iain Stewart. During the stay in both

neighbourhoods, participants were asked to take detailed field notes.

Focus Groups On the back of the field visit designed to take the geoscience researchers to the edge of their academic 'comfort zone', the authors facilitated two 90-minutes focus group sessions to explore the perceptions and attitudes of the ALErT investigators to the prospect of communicating their science more broadly. The initial focus group discussion took place on the afternoon of the field visit and and aimed to provide the participants a framework to reflect the field experiences and to voice their own individual views and concerns about their potential roles and responsibilities as geo-communicators. There then followed a 5-day technical field course along the North Anatolian Fault during which the participants were encouraged to have informal discussions among themselves about the broader issues of geo-communication. At the end of the field school, a second focus group was organised to elucidate the groups reflections on effective approaches of communicating hazard science to communities at risk and for community engagement. Both groups were structured around a set of predetermined questions, but the discussion was free-flowing. Both focus groups were moderated by the workshop facilitator Prof. Iain Stewart, who ensured that the variety of different ideas and opinions from as many group participants as possible could be voiced. He also facilitated an inventory of the discussion positions of the group, which were later structured on a flip-chart. The following paragraphs of section 4 will outline the key aspects that emerged through the interviews and group discussions among workshop participants and inhabitants. These summarize local perspectives on the mitigation process relevant to consider when developing communication strategies for (and with) at-risk communities. Section 5 will discuss the topics and concerns that emerged through the two focus group-approaches and summarize the recommendations elaborated by the workshop group.

I like the fact that some of the interviewees' statements appear in the bulk of the paper itself. However, here again I see a transparency issue. The reader cannot judge if the chosen pieces are representative of the whole sample. For sure you mention additional

testimonies to make your point: for example, page 9, lines 25-30; page 12, lines 14-21; page 15, lines 14-25; page 16, lines 7-13; or page 17, lines 19-25. I would like to see those statements as well. Maybe in a supplementary material file that could be accessed in the online version of the paper if you think that a table directly in the paper is too much.

Good point. In order to ensure a maximum transparency, we will provide a detailed overview of all the transcriptions in the appendix of the paper.

Also related to the interviewees' statements, I found that they are not always very well chosen. For example, page 8 in relation to participation. The statement that is summarized in the 26 seems to be a better highlight of the complaints than the one provided in line 23-24.

We agree with this comment and have changed the quotes here. The text now reads as follows:

A persistent complaint among interviewees related to a lack of citizen participation in the risk mitigation process, highlighting few, if any, established forums for science-public exchange, and an absence of contributions of local communities in the planning process and in regeneration activities.

"We found out about the [Urban Renewal] process only through their [the municipalities] marketing campaigns and the actual demolitions. And first hand experience. They never ask the public, they just construct a situation where it's all about them and their gains. There weren't any plans made in cooperation with the public." (Ali, Okmeydani)

The marketing campaigns that were initiated by the municipality were described as showcases aiming to create public acceptance for mitigation measures rather than as opportunities for open, critical disscussion. Inhabitants of Okmeydani criticized the one-sided orientation of these events, where the panel guests "talked the entire time and wouldn't allow us to ask critical questions".

Page 12, line 2: "four principal areas of concerns", what were the others ones? Explicitly mention why you chose to present only those.

We think this is a problem of wording, as the four categories encompass all topics and concerns raised in the focus groups. Therefore, we rephrased the sentence to:

In the following four paragraphs, we report, through the direct comments of the ALErT researchers, the areas of concern that emerged in the two focus group discussions following the urban field visit. The method of focus groups (as indicated in section 4) was chosen to gain an increased understanding on how participants reflect the experiences gained through the field visits, their role and responsibility in communicating to at-risk communities, and their recommendation of novel methods and approaches to seismic risk communication.

Page 12, line 18: here by "participants" do you refer to only the Turkish ones or the whole group? I would also have liked to know which of the researchers are Turkish or not. As you mention, the cultural aspect is important, so it would be interesting to see if there are perception's differences between the participants in relation to this point and also to see some discussion about that.

By refering to "participants", we refer to the whole group. Yet, we agree that a reflection of the impacts of cultural interpretations is a prerequisite to analyse risk communication in transdisciplinary formats. We have revised this section and provided a more in depth-discussion.

On the same line, I would have expected a deeper discussion section as well as greater use of the literature on some of the issues brought up by the participants, e.g. cultural aspects (page 13, lines 16-23) or the role of the geoscientists in communication (pages 14, lines 31-32).

This was done accordings to the reviewers suggestion. Also, relevant literature was consulted to contextualize the discussion with already existing reflections on the the

cultural side of risk communication and geoscientists perception of their role in the risk communication process.

Additionally, you do not challenge some points: e.g. "strict allocation of time spent on communication activities was seen as unrealistic". You certainly know that outreach activities are compulsory for some Horizon2020 projects and thus time must be allocated to those.

We have included these novel policies of the Horizon 2020 Frameworks for outreach and engagement in the section that outlines the increased attention paid to inter- and transdisciplinary approaches in risk communication.

Page 18, lines 23-25: provide references.

Page 19, section 6.3: Examples of universities proposing a joint technical and science communication master could be added as a good practice.

We appreciated this recommendation and provided information on existing joint MSc programmes: The section now reads as follows:

On a European level, a series of joint MSc-programmes already have reacted to this need of linking transdisciplinary methods, science communication training and disaster risk research. Examples are the MSc-programm "Risk Prevention and Disaster Management" at the University of Vienna, or "Disaster Risk Management and Climate Change Adaptation" at Lund University. Both programmes are based on close collaboration with practioners from governmental and non-governmental institutions. The joint masters of "Disaster Management and Risk Management" (University of Bonn/Federal Office for Citizen Protection and Disaster Support (BBK)) or "Geography of Environmental Risks and Human Security" (The United Nations University/the University of Bonn) are other examples of education initatives that are built on inter- and transdisciplinary frameworks.

Technical corrections: Page 6, line 22: one "for" too much. Page 15, line 25: "in order t

It was"

This was changed as per the reviewers suggestion.

---

## Short Comment (SC2) · 22 Mar 2016

The comment was uploaded in the form of a supplement:
http://www.nat-hazards-earth-syst-sci-discuss.net/nhess-2016-26/nhess-2016-26-SC2-supplement.pdf